# How to Get More Silver? Culture Media Adjustment Targeting Surge of Silver Nanoparticle Penetration in Apricot Tissue during in Vitro Micropropagation

Cristian Pérez-Caselles [1], Nuria Alburquerque [1], Lydia Faize [1], Nina Bogdanchikova [2], Juan Carlos García-Ramos [3], Ana G. Rodríguez-Hernández [4], Alexey Pestryakov [5] and Lorenzo Burgos [1,*]

[1] Group of Fruit Tree Biotechnology, Department of Plant Breeding, CEBAS-CSIC, Campus de Espinardo, Edif. 25, 30100 Murcia, Spain
[2] Centro de Nanociencias y Nanotecnología, Universidad Nacional Autónoma de México, Ensenada 22860, Mexico
[3] Escuela de Ciencias de la Salud, Campus Ensenada, Universidad Autónoma de Baja California (UABC), Ensenada 22890, Mexico
[4] Center for Nanosciences and Nanotechnology, Universidad Nacional Autónoma de México, CATEDRA CONACyT Researcher at CNyN–UNAM, Ensenada, B.C, Mexico City 04510, Mexico
[5] Research School of Chemistry and Applied Biomedical Sciences, Tomsk Polytechnic University, 634050 Tomsk, Russia
* Correspondence: burgos@cebas.csic.es; Tel.: +34-968396200 (ext. 445317)

**Abstract:** The use of silver nanoparticles (AgNPs) is increasing nowadays due to their applications against phytopathogens. Temporary Immersion Systems (TIS) allow the micropropagation of plants in liquid media. This work aims to develop an effective protocol for apricot micropropagation in TIS and to study the necessary conditions to introduce AgNPs in apricot plants, as well as the effect of its application on proliferation-related parameters. AgNPs were introduced in different media at a concentration of 100 mg L$^{-1}$ to test the incorporation of silver to plant tissues. Silver content analysis was made by Inductively Coupled Plasma-Optical Emission Spectroscopy (ICP-OES). The effect of initial shoot density and the addition of AgNPs on micropropagation were evaluated after four weeks in culture on TIS. Productivity, proliferation, shoot-length and leave surface were measured. The better micropropagation rate was obtained with 40 initial shoots, 2 min of immersion every 6 h and 3 min of aeration every 3 h. To introduce AgNPs in apricot plants it is necessary to culture them in liquid media without chloride in its composition. These results will contribute to the development of an in vitro protocol for virus inhibition by AgNPs application. This depends on the introduction of Ag nanoparticles within the plant tissues, and this is not possible if AgNPs after interaction with Cl$^-$ ions precipitate as silver chloride salts.

**Keywords:** TIS; AgNPs; liquid media; virucide; micropropagation

## 1. Introduction

Emerging and re-emerging viruses are considered a continuous threat to human, animal and plant health as they are able to adapt to their current host, migrate to new hosts, as well as develop strategies to escape antiviral measures [1–3].

Diseases caused by plant viruses can cause severe damage in crops and the magnitude of losses in crop yield or product quality can be devastating [4]. Among control strategies for the protection of plants against viral diseases, using pesticides against virus vectors is the most common [5]. However, due to environmental damages caused by an excessive use of pesticides, it is necessary to fight plant diseases without harming natural ecosystems [5,6]. Nanotechnology may be a useful tool in protecting crops against viral diseases [7].

Silver nanoparticles (AgNPs) have been used because of their antimicrobial activity. They are able to control different types of pathogens such as bacteria, virus and fungus,

without affecting the development and functionality of plants [8,9]. Additionally, it has been described that they may improve, in vitro, the growing of plants when added to the media [10,11].

Apricot (*Prunus armeniaca* L.) is one of the most economically important species of stone fruits. World production in 2020 was 3.7 million tons in 562.475 hectares [12]. Apricots have been traditionally cultured in Spain because in this country, there are many areas with appropriate environmental conditions for cultivation of different apricot cultivars.

In vitro propagation allows for the production of abundant clonal plant material and has been used for germplasm conservation in controlled and aseptic conditions. The fruit biotechnology group at CEBAS-CSIC, Murcia (Spain) has developed efficient protocols for the micropropagation of apricot cultivars in semisolid media [13]. Additionally, there is the possibility of propagating plant material in liquid media by using Temporal Immersion Systems (TIS). These are bioreactors where filtered air can be introduced to displace the liquid medium to the part of the reactor where the plant material is placed and therefore immersing it for a certain time. Later air can be introduced to return the liquid medium to its original location or just to dry out the plants to avoid excessive humidity [14]. By optimizing immersion and aeration times plants can be proliferated without producing detrimental physiological changes. TIS allows for the introduction, transport and accumulation of specific compounds, such as AgNPs, in plants [15].

To the best of our knowledge, AgNPs have been used to control plant viruses by spraying leaves [16–18] but have never been added to culture media during in vitro propagation to study their effect on virus or other pathogens within the plants. Here, we have used Argovit®, a commercial product of AgNPs with an antimicrobial activity previously demonstrated [19]. It was also revealed the virucidal properties of Argovit® AgNPs in different biological systems: infectious bronchitis coronavirus in chickens [20], *canine distemper virus* in dogs [21], white spot syndrome virus in shrimp [22], Rift Valley fever virus in vitro and in vivo [23] and SARS-CoV-2 coronavirus in vitro and in health workers [24].

The objectives of this work are: (1) to develop the micropropagation protocol of apricot in TIS by searching the conditions providing penetration of Argovit® AgNPs into the tissues of in vitro propagated apricot plants by adding AgNPs to the media and (2) to determine if the addition of AgNPs has any effect on the apricot proliferation. This work will serve as the first stage for a long-term objective which is to produce virus and viroid-free apricot plants during micropropagation. For this long term objective, the penetration of AgNPs with antiviral properties into plan tissue is essential, which is the short term objective of the present publication.

## 2. Materials and Methods

### 2.1. Plan Material, Maintenance, and Proliferation in Semisolid or Liquid Medium

The apricot cultivars 'Canino' and 'Mirlo Rojo' are part of the apricot germplasm maintained in vitro at the Fruit Biotechnology Group laboratory in CEBAS-CSIC and were used for the experiments in this work. In vitro plants are maintained in a culture chamber at temperature of $23 \pm 1$ °C, 16/8 h light/dark photoperiod (56 $\mu$mol m$^{-2}$s$^{-1}$ of light intensity) and are transferred to fresh medium every four weeks.

Micro-shoots grow and multiply in 500 mL glass jars, 12 shoots per jar with initial approximated length of 1.5 cm (Figure 1). The semisolid culture medium (SSM) used in this work has been described by Wang et al. [25]. Basal salt composition can be found in Table 1 but briefly SSM has QL [26] macronutrients, DKW [27] micronutrients and vitamins plus the addition of 3 mM calcium chloride, 0.8 mM phloroglucinol, 3% sucrose, 0.7% agar, 1.12 mM 6-benzylaminopurine riboside (BAPr), 0.05 mM IBA, 2.1 mM Meta-Topolin (6-(3-hydroxybenzylamino) purine) and 29.6 mM adenine. Medium pH is adjusted to 5.7 by adding NaOH 0.1 N before autoclaving at 121 °C for 20 min.

**Figure 1.** Temporal Immersion System (SIT) in Plantform$^{TM}$ bioreactors. Initial stage of 'Canino' shoots (**A**,**B**), and proliferation after four weeks (**C**,**D**).

**Table 1.** Culture media used for apricot micropropagation and modifications to study AgNPs incorporation in plant tissues.

| Media Codes | Basal Salts | CaCl$_2$ (3 mM) | Other Components [z] | Agar (7 gL$^{-1}$) |
|---|---|---|---|---|
| SSM | QL + DKW | Yes | a | Yes |
| LM1 | QL + DKW | Yes | a | No |
| LM2 | QL + DKW | No | b | No |
| LM3 | QL + DKW | No | a | No |
| LM4 | QL modified [y] + DKW | No | a | No |

[y] QL macronutrients was modified by adding 1,908 mg L$^{-1}$ Ca(NO$_3$)$_2$ · 4 H$_2$O and 160 mg L$^{-1}$ NH$_4$NO$_3$ instead of 1,200 mg L$^{-1}$ and 400 mg L$^{-1}$, respectively. [z] a: 0.8 mM phloroglucinol, 3% sucrose, 1.12 mM BAPr, 0.05 mM IBA, 2.1 mM Meta-Topolin and 29.6 mM adenine; b: 3% sucrose, 1.96 mM BAP and 0.2 mM IBA.

For apricot micropropagation in liquid media 4 L Plantform$^{TM}$ bioreactors were used. To maintain sterility inside the bioreactor after autoclaving, all manipulation was performed inside the hood and pressure air was introduced through 0.22 μm air filters. Liquid medium with the same composition as SSM was used, but without agar (LM1 in Table 1).

In each bioreactor, 500 mL of LM1 was poured and every four weeks, the medium was refreshed. Based on the results from previous experiments, the following optimal protocol was applied: 2 min of immersion every 6 h and 3 min of aeration every 3 h. To improve micropropagation in bioreactors with liquid media the effect of initial density of shoots was evaluated starting the culture with 10, 20, 30 or 40 'Canino' shoots.

## 2.2. Physicochemical Characteristics of AgNPs

Argovit$^®$-7 AgNPs used in this work were kindly donated by the Scientific-Production Centre Vector-Vita Ltd., Novosibirsk, Russia. According to the manufacturer's data Argovit-7 is an aqueous solution with 1.2% (*w/w*) of metallic silver and 18.8% (*w/w*) of polyvinylpirrolidone (PVP), with integral AgNPs concentration (metallic silver and PVP) of 200 g L$^{-1}$.

High-Resolution Transmission Electron Microscope (HR-TEM) a JEM-2010 (JEOL©) was used to determine AgNPs morphology. NPs average diameter (ϕAg) and size distribution analysis was determined from TEM micrographs (n = 150) using ImageJ Software [28]. Thermogravimetric analysis (TGA) and differential scanning calorimetry (DSC) measurements were performed with a DSC/TGA thermal analysis system (SDT Q600, Newtown (PA), USA) under argon atmosphere from 30 to 750 °C at a rate of 10 °C min$^{-1}$. Surface plasmon analysis was performed using AgNPs dilutions on distilled water acquired in a Thermo-Scientific Genesys 30 UV/Vis spectrophotometer with a 1 cm optical path quartz cell. Zeta potential and hydrodynamic diameter were determined by dynamic light scattering (DLS) using a Zetasizer Nano-ZS with capillary cuvettes (Malvern Instruments$^®$, Malvern, UK).

## 2.3. Addition of AgNPs to Different Media Compositions

To test the incorporation of silver to plant tissues, AgNPs were introduced in different media at a concentration of 100 mg L$^{-1}$ (metallic silver plus PVP). AgNPs were added after autoclaving and cooling down the media in the semisolid (SSM) and liquid medium (LM1) with a similar composition. Additionally, other basal salt combinations were tested as well

as modifications of LM1. The media tested are detailed in Table 1 but briefly medium LM2 is based in QL [26] macronutrients and DKV [27] micronutrients and vitamins, 3% sucrose, 1.96 mM BAP and 0.2 mM IBA. LM3 is the medium LM1 without calcium chloride. Finally, medium LM4 is LM1 without calcium chloride, but to compensate for a possible calcium deficiency calcium nitrate was added and ammonium nitrate was reduced to avoid excessive nitrogen in the medium (Table 1).

### 2.4. Micropropagation Evaluation

The effect of initial shoot density and the addition of AgNPs on micropropagation were evaluated after four weeks in culture. Recorded dependent variables were proliferation (number of new growths longer than 1 cm from each initial shoot), mean length of new shoots longer than 1 cm, productivity (the product of proliferation and mean new shoot length) and leaf area (which was measured for the fourth leaf on the main shoot). Twenty shoots from each treatment were measured.

### 2.5. Determination of Silver Content in the Plant Tissues

To determine silver content 200 mg of dry material, obtained from whole shoots without basal calli, were used. Fresh samples were dried out at 60 °C for 72 h. Dried powder was subjected to acid digestion in a $HNO_3/H_2O_2$ solution (4:1) using an UltraCLAVE microwave. Silver concentration was determined by Inductively Coupled Plasma-Optical Emission Spectroscopy (ICP-OES) at CEBAS Ionomic Service using iCAP 7000 (Thermo Fisher Scientific, Waltham, MA, USA). Each silver content is the average from three values obtained for independent samples.

### 2.6. Statistical Analyses

Results were analyzed using SAS version 9.4 (SAS Institute, Cary, NC, USA). A variance analysis determined if there were significant differences between treatments (initial shoot density or media). The effect of the initial shoot density was evaluated by a trend analysis and orthogonal contrasts.

Data of silver content in plant material were transformed with the natural logarithm before the analyses to homogenize variances. Results obtained from each culture medium were compared with the control, SSM without AgNPs addition, by means of one-tail Dunnett test.

Proliferation variables evaluated for plants grown in media without calcium chloride were compared after the ANOVA with a Duncan post-hoc test.

## 3. Results

### 3.1. Physicochemical Characterization

AgNPs formulation used in this work was characterized by TEM, UV-is, DCS/TGA and DLS analysis. TEM images exhibit spherical AgNPs, size distribution range between 22.1 and 44.8 nm, with an average size of 31.9 nm, and with tendency of these nanoparticles to agglomerates into groups with the size more than 100 nm, as shown in Figure 2A. UV-vis spectra show three absorption maxima at 407, 439 and 526 nm, associated with the size distribution of the sample (Figure 2B). DSC/TGA analysis allows for the determination of the composition of the AgNPs formulation, with 75.82% of water (loss at 85 °C), 22.21% of K-17 polyvinylpyrrolidone (PVP), and 1.96% of metallic silver. The sharp endothermic process found at 447 °C let a pure coating agent be identified (Figure 2C). Hydrodynamic diameter and zeta potential measured wit DLS are 56.1 nm and −3.85 mV, respectively.

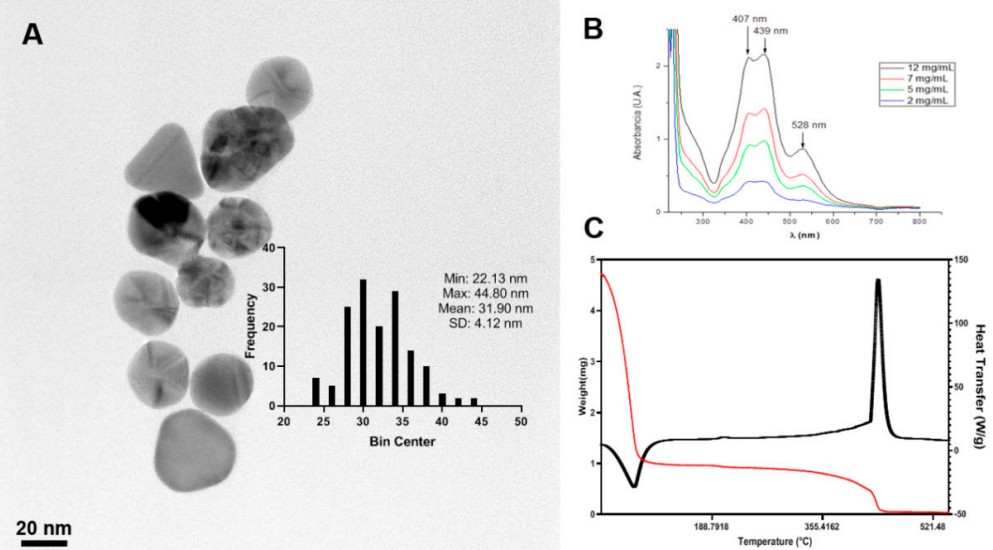

**Figure 2.** Physicochemical properties of AgNPs. (**A**) TEM images showing AgNPs morphology and size distribution (inset histogram), (**B**) surface plasmon registered at different concentrations in distilled water, and (**C**) DSC/TGA analysis showing the mass loss as a function of temperature increase (red line) and the endothermic/exothermic processes (black line).

### 3.2. Effect of Initial Shoot Density

The number of shoots in the bioreactor at the beginning of the culture cycle affects different parameters related to micropropagation in the TIS. 'Canino' micropropagation was studied with different initial shoot density (10, 20, 30 and 40 shoots). Figure 3 shows a trend analysis for the micropropagation variables with the increase in initial shoot density in TIS. Moreover, a prediction equation was calculated for those variables where significant differences between treatments were found.

Statistical analyses showed significant differences ($p < 0.01$) in productivity which rises with the increase in the initial shoot number (Figure 3). No significant differences were found for the number of new shoots longer than 1 cm when the initial shoot density increased. However, mean shoot length was significantly affected ($p < 0.001$) with a maximum mean length of 2.26 cm for the highest initial shoot density (40 shoots). Leaf area was also significantly affected ($p < 0.001$) by initial density and drastically increased, reaching a maximum value of 2.62 cm$^2$ at the highest initial shoot number (Figure 3). Considering all parameters studied, the best results were obtained with initial shoot number of 40 shoots, and this was the shoot density in following experiments. With the same immersion and aeration conditions and an initial density of 40 shoots we tested the effect of TIS on 'Mirlo Rojo' (Table 2, medium LM1 without AgNPs). Therefore, 'Mirlo Rojo' had lower proliferation rates than 'Canino' in these culture conditions.

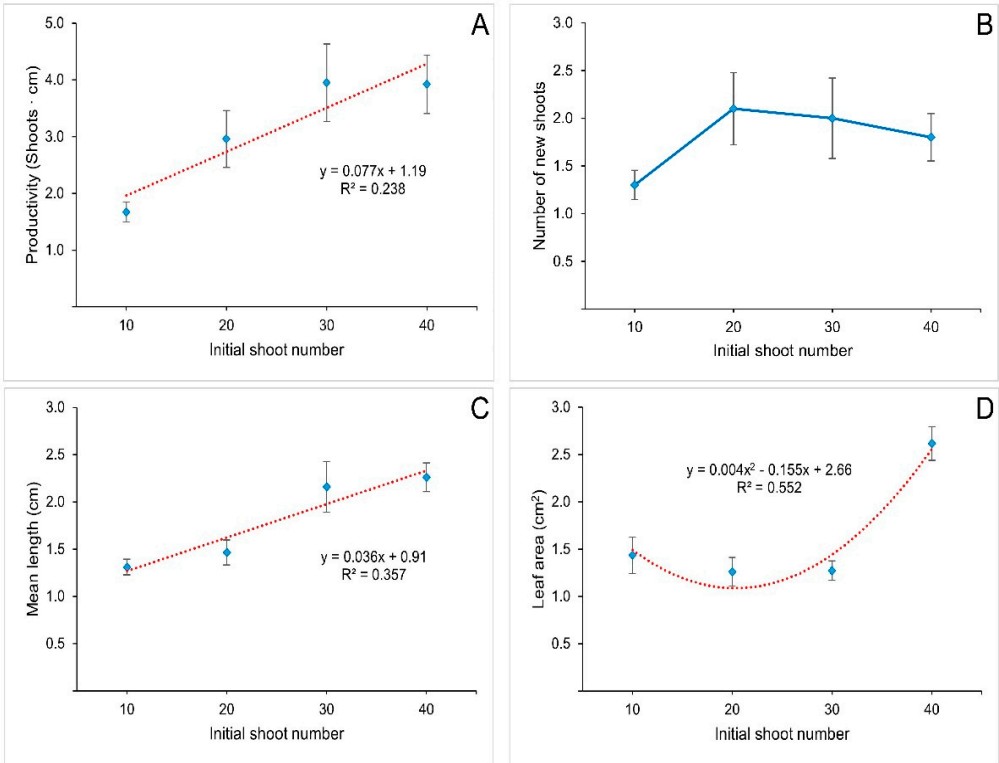

**Figure 3.** Effect of initial shoot density in TIS on productivity (**A**), number of new shoots (**B**), mean length (**C**) and leaf surface (**D**) of 'Canino' shoots. Discontinuous line represents the results of trend analysis. Bars are the standard error of mean values.

*3.3. Silver Content within the Plants*

The long-term objective of this study is to study virucidal activity of AgNPs in plant tissue. It is paramount that AgNPs penetrate the plant tissues. Therefore, the effect of media composition containing 100 mg $L^{-1}$ AgNPs (Table 1) on the silver content in the plant tissue after four weeks of cultivation was studied. Figure 4 shows silver content found in 'Mirlo Rojo' shoots cultured in each of the media described in Table 1. Additionally, silver content in 'Canino' shoots cultured in control, SMM and LM3 media are shown.

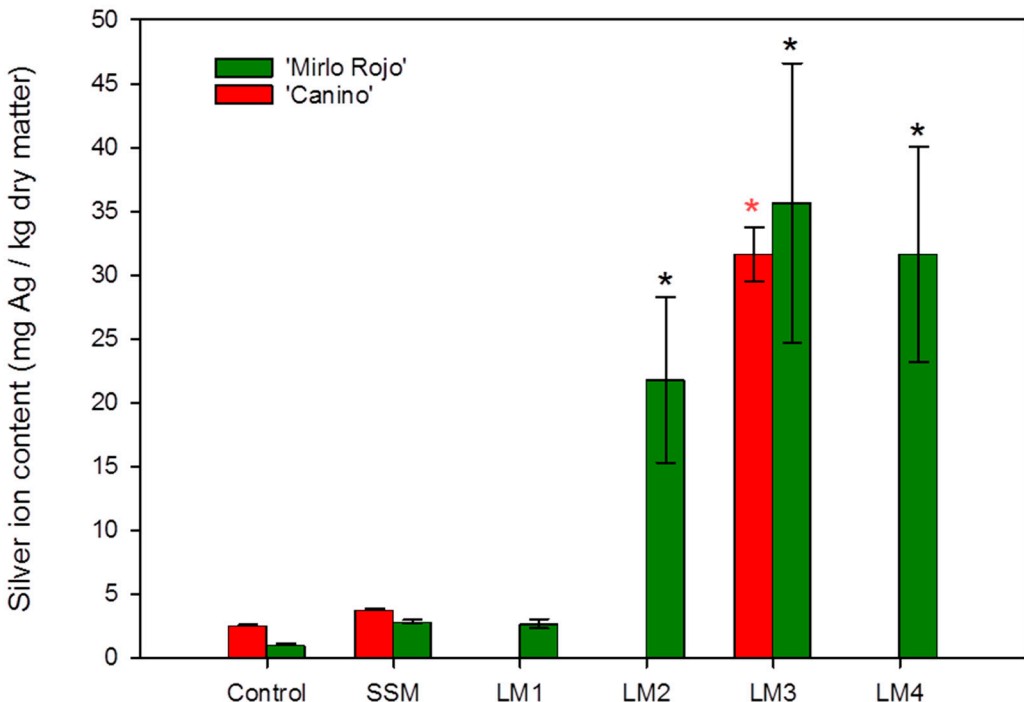

**Figure 4.** Silver concentration in 'Mirlo Rojo' and 'Canino' shoots cultured in different media containing 100 mg L$^{-1}$ AgNPs. Control medium is SSM without AgNPs. Other media contents are described in Table 1. Bars represent standard errors. Asterisks indicate treatments significantly different from the controls for each cultivar ($p < 0.05$) according to a one-tail Dunnett test.

Control treatment corresponds to the shoots cultured in semisolid medium without AgNPs addition (Figure 4). SSM (containing agar and CaCl$_2$) and LM1 (without agar and with CaCl$_2$) with AgNPs showed a slight, but not significant, increase in silver concentration compared with control. However, for LM2, LM3 and LM4 (all of them without agar and CaCl$_2$) a significant increase ($p < 0.05$) in the silver content compared with the control was recorded. LM3 with 35.6 mg of silver per kg of dry plant was the medium where the highest content of silver was found (Figure 4). When the silver concentration was studied in 'Canino' shoots cultured in SMM and in LM3, 3.71 and 31.6 mg/kg of silver were found, respectively. These are values similar to those found for 'Mirlo Rojo' in these same media. Only silver content in LM3 significantly differed from the control of 'Canino' shoots cultured on SMM without AgNPs.

*3.4. Effects of AgNPs on Apricot Micropropagation*

For the short- and long-term objectives of this work, it is essential to determine if the addition of AgNPs has any detrimental effect on the micropropagation of apricot cultivars. Therefore, different variables related to micropropagation were measured after exposure of 'Mirlo Rojo' apricot shoots to media with AgNPs. Since the effect of initial number of shoots, proliferation, mean shoot length, productivity and leaf area (Figure 3) were studied without AgNPs, the same micropropagation variables were measured with AgNPs at the concentration of 100 mg L$^{-1}$ after culturing 'Mirlo Rojo' shoots in LM1 (control medium with CaCl$_2$) and LM2, LM3 and LM4 (without CaCl$_2$) media (Table 2).

A specific contrast was performed in order to determine the effect of AgNPs addition on micropropagation of 'Mirlo Rojo' by comparing LM1 with the average of values recorded in the rest of liquid media with AgNPs (Table 2, treatment LM is the average of all media containing AgNPs). Treatments with AgNPs increased productivity when compared with the control LM1 (Table 2) due mainly to an increase in the number of new shoots (proliferation). On the other hand, there were not significant differences in mean shoot

length. Leaf area was significantly larger in the media with AgNPs than in the control medium (Table 2).

Since the addition of AgNPs had a significant effect improving productivity we decided to study which composition of liquid media can provide the best micropropagation-related variables for 'Mirlo Rojo' shoots (Table 2). ANOVA found significant differences among the three AgNPs-containing media (LM2, LM3 and LM4) ($p < 0.05$) for all variables studied. Productivity and the number of new shoots were significantly better in media LM3, but mean shoot length and leaf area were larger in LM2 (Table 2).

**Table 2.** Effect of AgNPs addition on micropropagation of 'Mirlo Rojo' apricot in different liquid media.

| Treatment [z] | Productivity Shoots x cm | Proliferation Shoot Number | Shoot Length cm | Leaf Area cm$^2$ |
|---|---|---|---|---|
| LM1 (no AgNPs) | 1.92 ± 0.41 | 1.40 ± 0.22 | 1.35 ± 0.11 | 2.53 ± 0.17 |
| LM (+AgNPs) | 4.95 ± 0.34 * | 3.57 ± 0.27 * | 1.50 ± 0.10 | 3.79 ± 0.50 * |
| LM2 (+ AgNPs) | 5.33 ± 0.65 [ab] | 3.08 ± 0.48 [b] | 2.06 ± 0.27 [a] | 7.21 ± 1.16 [a] |
| LM3 (+ AgNPs) | 6.10 ± 0.61 [a] | 4.58 ± 0.48 [a] | 1.35 ± 0.05 [b] | 2.50 ± 0.31 [b] |
| LM4 (+ AgNPs) | 3.92 ± 0.46 [b] | 3.22 ± 0.38 [b] | 1.23 ± 0.04 [b] | 2.38 ± 0.33 [b] |

[z] Asterisks represent significant differences ($p < 0.05$) between the media without AgNPs addition (LM1) and the average of AgNPs-containing media (LM+AgNPs) according to a specific contrast. Different letters within the AgNPs-containing media represent significant differences ($p < 0.05$) between treatments according to a Duncan test.

The best medium found for 'Mirlo Rojo' (LM3) was tested for 'Canino' that produced a ratio of 6.1 ± 1.36 shoots with an average length of 1.51 ± 0.34 and leaf area of 2.57 ± 0.57, which gives a productivity of 9.12 ± 2.04 even higher than in 'Mirlo Rojo'.

## 4. Discussion

The use of AgNPs is increasing due to their ability to eliminate phytopathogens, bacteria, virus and fungus. AgNPs have been successfully used to disinfect explants when establishing plants in vitro [29,30]. The antivirus efficiency of Argovit@ AgNPs has been demonstrated in different biological systems. Bogdanchikova et al. (2016) described that, after application of Argovit AgNPs, symptoms of *Canine distemper virus* (CDV) disappeared in dogs without neurological symptoms of the disease. Argovit® AgNPs were effective in the treatment or prophylaxis of infectious bronchitis coronavirus in chickens [20], white spot syndrome virus in shrimp [22,31], as well as in in vitro and in vivo models of Rift Valley fever virus [23] and SARS-CoV-2 coronavirus [24].

When applied on guar leaf surface along with *Sunhemp rosette virus* (SHRV), AgNPs have a preventive effect on the presence of symptoms on leaves (Jain & Kothari, 2014). Similarly, a post-infection treatment by spraying with AgNPs prevented the destructive symptoms caused by the *Bean yellow mosaic virus* (BYMV) in faba bean plants [17]. Recently, in the same species, leaf application of AgNPs have been demonstrated effective in preventing infection with BYMV and reducing virus replication [16]. Some authors found that AgNPs reduce infection ability of viruses when applied as a preventive treatment [32]. However, to the best of our knowledge, there is not a methodology for producing virus-free plants by adding AgNPs to the in vitro culture media.

The Fruit Biotechnology Group at CEBAS-CSIC has a collection of apricot cultivars maintained in vitro culture in semisolid medium. A protocol for the micropropagation of apricot cultivars was already available. Accordingly, 'Mirlo Rojo' shoots were cultured in SSM medium (Table 1) with the addition of AgNPs at 100 mg L$^{-1}$ (silver and PVP) to study the capacity of AgNPs to penetrate into plant tissues. Bello-Bello et al. [11] described that using Argovit® between 50 and 100 mg L$^{-1}$ allowed for the micropropagation of sugar cane (*Saccharum spp*) but resulted toxic at concentrations over 200 mg L$^{-1}$. Therefore, we decided to choose 100 mg L$^{-1}$ Argovit-7 AgNPs concentration since the objective was to use the highest possible concentration without detrimental effect on micropropagation. After four

weeks of culture in SSM, silver was not detected in 'Mirlo Rojo' or 'Canino' shoot tissues (Figure 4). This may be due to slow migration of AgNPs inside the agar matrix. Agar at concentration used in this work (7 g L$^{-1}$) have a pore size of around 100 nm [33]. This could be close to 56.1 nm of hydrodynamic diameter of AgNPs nanoparticles studied here which would impede the mobility of nanoparticles and therefore the possibility to be absorbed by the plant. Moreover, the results of TEM study showed that AgNPs agglomerates into groups with a size more of than 100 nm that does not allow their diffusion through 100 nm agar pores. Apparently low negative charge of the particles (zeta potential is −3.85 mV) facilitate agglomeration of AgNPs into the groups.

A methodological approach to address this problem could be application of liquid medium to culture apricot shoots. Castro-González et al. [15] found that silver from Argovit$^{@}$ formulation was transported from culture medium and was accumulated within plant tissues when Stevia (*Stevia rebaudiana* B.) was cultured in liquid medium in TIS. Therefore, conditions including the application of liquid medium to culture apricot in TIS were also applied in the present work.

There are several studies on the micropropagation of woody plants in TIS [14,34]. However, it is necessary to modify protocols for each species, and even for each cultivar.

The main parameters affecting efficiency of micropropagation in TIS are immersion time, aeration time, bioreactor volume and culture medium volume [35]. Immersion and aeration times determine nutrient absorption and control of hyperhydration [35]. Hyperhydration is one of the main limiting factors of micropropagation in TIS due to the apparition of physiological and morphological disorders, such as irregular shoots growth or production of anomalous leaves [36]. Our previous experience with apricot revealed that this species is very susceptible to hyperhydration [37] and therefore short immersion times and frequent aeration periods are necessary. These results showed that a combination of 2 min of immersion every 6 h and 3 min of aeration every 3 h leaded to effective proliferation of apricot in TIS.

Vegetative propagation in bioreactors is directly related to the bioreactor volume. Propagation of grapes in bioreactors with different volumes demonstrated that the larger volume the higher number of longer new shoots [38]. In the present work we found that increased number of initial shoots had a positive effect in apricot micropropagation. Figure 3 shows that productivity of 'Canino' raised with the increase in the initial shoot number. This effect could be explained by the beneficial conditions effect of our experiments (limited space, Figure 1C,D), with shoots located close to each other, which can stimulate vertical growth of the main stems. When there is a large space between shoots, they tend to grow horizontally which limits their length.

Medium volume depends on the type of bioreactor used. To produce efficient plant propagation in TIS, medium should completely cover the explants during the immersion phase. For the 4 L Plantform$^{TM}$ bioreactors used here, 500 mL of medium gave good results immersing completely the explants.

Culture conditions were established for 'Canino' in TIS because its behavior in semisolid medium is well known. The best determined conditions in TIS were tested for 'Mirlo Rojo'. We found expected differences between genotypes but both cultivars were successfully propagated in TIS. We paid special attention to 'Mirlo Rojo' in the experiments with addition of AgNPs because we have this cultivar infected with *Hop stunt Viroid*.

'Mirlo Rojo' shoots were cultured in liquid medium LM1 supplemented with 100 mg L$^{-1}$ AgNPs. Results indicate that silver did not penetrate the plant tissues (Figure 4). There was a slight non-significant increase, as compared with the control medium without AgNPs, but it could be due to silver precipitated on the shoots surface. At the end of the culture cycle, we observed solid precipitate at the bottom of the bioreactors, probably a precipitated salt. Major and minor nutrients for the plant in culture media are in the form of salts that are dissociated at the pH of the media and then plants absorb the dissociated ions. AgNPs may react with different salts components of the media. For instance, chloride, sulfate, or nitrate anions can react with silver cations (which slowly are

released from AgNPs) forming salts that have different degrees of solubility. We decided to check if chloride anions, dissociated from CaCl$_2$, (a major component of SSM and therefore of LM1) can react with Ag$^+$ forming the highly insoluble silver chloride salt [39]. Although we found precipitate at the bottom of the bioreactors after using media without calcium chloride, it was in much lower amounts than in medium with CaCl$_2$, which supports the hypothesis that silver may precipitate in the same way after reacting with other salts but in less amounts than with CaCl$_2$. Modifying culture media is always a risk since they usually are very well balanced, and the modification of any component may strongly affect micropropagation rates and plant behavior in vitro. Nevertheless, we decided to start testing modifications. The choice fell on CaCl$_2$ since it is in larger amounts than sulfates or nitrates and solubility of AgCl is much lower than the other silver salts. On one hand LM2 medium formulation is frequently used, after adding agar, to micropropagate some apricot cultivars, as a semisolid medium. Moreover, on the other hand, LM2 does not have CaCl$_2$ in its composition. Therefore, we included it, just as a liquid medium without modifications. Additionally, we completely removed CaCl$_2$ in LM3. In LM4, CaCl$_2$ was also removed substituting Ca cation by Ca(NO3)$_2$ and reducing NH$_4$NO$_3$ to compensate for a possible calcium deficiency and avoid excess of nitrogen, respectively.

After culturing 'Mirlo Rojo' shoots in three different liquid media without chloride ions (LM2, LM3 and LM4) and with the addition of 100 mg L$^{-1}$ AgNPs, silver was found in plant tissues from the shoots cultured in these three media in concentrations significantly higher than in shoots cultured in the control medium (Figure 4). LM3 promoted the penetration of the highest silver concentration to shoot tissues from culture medium compared with LM2 and LM4. Silver introduction in 'Canino' shoots cultured in LM3 was also significantly larger than in control shoots. Therefore, culture in liquid medium with a simple change consisting in removing CaCl$_2$ avoided immobilization of AgNPs in agar matrix and precipitation of silver chloride salts. These modifications facilitated the penetration of silver into plant tissue. This is the first, inevitable and therefore paramount, step ensuring the interaction of AgNPs with pathogenic microorganisms inside shoot tissues.

As indicated above, the modification of culture media may be risky and, therefore, the influence of AgNPs addition to media needs to be evaluated to discard a negative effect on apricot micropropagation. Table 2 shows a comparison between micropropagation parameters of 'Mirlo Rojo' shoots in LM1, liquid medium whithout AgNPs addition, with those in LM2, LM3 and LM4, where AgNPs were added, and silver is entering plant tissues. A positive effect of AgNPs addition on 'Mirlo Rojo' and 'Canino' micropropagation was found, increasing productivity, mainly due to a higher number of new shoots. This positive effect was also described in sugarcane micropropagation [11], vanilla (*Vanilla planifolia* Jacks. ex Andrews) [10] and stevia [15] when using Argovit$^®$ in TIS. Biostimulants are compounds that induce a physiological response in the plant improving growth and development [40]. AgNPs have a biostimulation effect when they are used in TIS.

In order to maximize the efficiency of AgNPs addition on apricot micropropagation it is necessary to use the optimal medium composition. From the media tested here, LM2 and LM3 lead to higher productivity. In LM3 the improvement was due to the production of a higher number of new shoots and a higher proliferation, than in the other two media without calcium chloride, whereas LM2 induced the production of significantly longer shoots with a larger leaf surface (Table 2). It should be noted that high leaf surface leads to dominant use of energy to produce biomass instead of shoot growth. When the effect of LM3 was tested on 'Canino' it also improved micropropagation rates. Therefore, LM3 medium, traditionally used for apricot cultivar micropropagation, after elimination of calcium chloride proved to be the most effective medium for AgNPs penetration into the plant tissue.

These results can undoubtedly be used in further studies targeting AgNPs application for pathogen microorganism growth inhibition in plant tissue during micropropagation in TIS.

## 5. Conclusions

The long-term objective of this work is the application of AgNPs treatments to prevent or cure microbe's contamination in plant tissues during micropropagation. However, problems related with immobilization of nanoparticles in the agar matrix and/or precipitation of silver salts prevented the migration of silver into plant tissues. By using liquid media in Temporal Immersion Systems and a simple medium modification consisting in removal of calcium chloride we have demonstrated that silver penetration into plant tissues increases, as average for both cultivars, 23-fold and without any detrimental effect on apricot productivity, that on the contrary, was improved by 2.8-fold as average for both cultivars, due to the addition of AgNPs in the culture media.

**Author Contributions:** Conceptualization, L.B.; methodology, L.F., C.P.-C., A.G.R.-H. and J.C.G.-R.; investigation, C.P.-C., N.A. and L.B.; resources, N.B. and A.P.; writing-original draft preparation, L.B. and C.P.-C.; writing-review and editing, N.B. and N.A.; supervision, L.B. All authors have read and agreed to the published version of the manuscript.

**Funding:** This research was funded by Project CARM-Agroalimentario Ref: CA20565 "OSIRIS".

**Institutional Review Board Statement:** Not applicable.

**Informed Consent Statement:** Not applicable.

**Data Availability Statement:** Not applicable.

**Acknowledgments:** Authors wish to thank Spanish Ministry of Education for a FPU fellowship granted to C.P.-C. and Russian Science Foundation and Tomsk region for grant 22-13-20032. The authors thank Amelia Olivas from CNyN-UNAM for TGA/DGS spectra acquisition.

**Conflicts of Interest:** The authors declare no conflict of interest.

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
