# Peer review of "How to Get More Silver? Culture Media Adjustment Targeting Surge of Silver Nanoparticle Penetration in Apricot Tissue during in Vitro Micropropagation"

_horticulturae, doi:10.3390/horticulturae8100855_

Round 1
Reviewer 1 Report
Thanks for the opportunity to review this research intitled “How to get more silver? Culture media adjustment targeting 2 surge of silver nanoparticle penetration in apricot tissue during 3 in vitro micropropagation”. The topic treated by the authors is very interesting and can be a promising first step towards the application of silver nanoparticles against plant viruses. The introduction is concise and explanatory; the chapter Materials and Methods explains well the methods applied. However, the results are lacking: the authors present data relating to shoot density only for the "Canino" cultivar and silver concentration and micropropagation parameters only for the "Mirlo Rojo" cultivar. Authors must include complete data for both cultivars in the publication.
I have some notes and questions for the authors:
1) insert in figure 1 the cultivar presents in the photos.
2) Insert in figure 3 the data about “Mirlo Rojo” cultivar
3) Insert in figure 4 and Table 2 the data about “Canino” cultivar
4) Line 315: is correct "Canine" rose?
5) References : write the genera and species correctly and in italics (10, 11, 14, 16, 21, 25, 29, 30)
Reviewer 2 Report
The manuscript “How to get more silver? Culture media adjustment targeting surge of silver nanoparticle penetration in apricot tissue during in vitro micropropagation” is dealing with the effect of Argovit® AgNPs applied in temporary immersion system. Authors reported significantly higher accumulation on silver in shoots after culture them in TIS in liquid media without chloride. By using Temporal Immersion Systems and a simple medium modification it is demonstrated that silver penetration into plant tissues increases without detrimental effect on apricot propagation. The results presented in manuscript has same merit and it is not acceptable for a publication in present form. The whole manuscript is not very well written and has several methodological deficiencies:
1. The whole study about the effect of Argovit® AgNPs was done with Mirlo Rajo cultivar? Why results about cultivation in TIS of Canino cultivar are included in this manuscript? There were no results about the effect of Argovit® AgNPs on this cultivar. It is not possible to compare results presented in Figure 3 and Table 2 since they are different cultivars (Line 232-235)
2. Line 228-230 What means sentence: ” Although the short- and long-term objectives of this work was not related to micropropagation, it is essential to determine if the addition of AgNPs has any detrimental effect on the micropropagation of apricot cultivars”. According to presented results this manuscript is about micropropagation of apricot in TIS.
3. It seems logical to present the effect of Argovit® AgNPs on micropropagation parameters and then Ag content in shoots?
4. Line 237-239: What means sentence: “A specific contrast was done to determine the effect of AgNPs addition on micropropagation of ‘Mirlo Rojo’ by comparing LM1 with the average of values recorded in the rest of liquid media with AgNPs”. Clarify.
5. Line 384-385 The short and long term objectives of the study should be stated in introduction section
Round 2
Reviewer 2 Report
The authors significantly improved first version of the manuscript.
Author Response
I do not see that the referee asked for additional changes. I only found that he/she considered that the introduction could be improved. It is difficult to know what to change without additional information. The introduction starts with a brief paragraph about the importance of viral diseases. Continues with the possibility of using AgNPs to control these pathogens. Then describes a little bit the importance of apricots and the possibilities to propagate them in vitro by using different systems.
Finally we reviewed all we have been able to find about the use of AgNPs to control virus diseases in plants and animals or insects and established the objectives of this manuscript.
Please if you have problems with the structure or any other thing let us know but without further explanation we do not see how to improve it